# UNSUPERVISED DETECTION OF CELL ASSEMBLIES WITH GRAPH NEURAL NETWORKS

**Roman Koshkin**
Neural Coding and Brain Computing Unit
Okinawa Institute of Science and Technology
Okinawa, Japan
`roman.koshkin@oist.jp`

**Tomoki Fukai**
Neural Coding and Brain Computing Unit
Okinawa Institute of Science and Technology
Okinawa, Japan
`tomoki.fukai@oist.jp`

## ABSTRACT

Cell assemblies, putative units of neural computation, manifest themselves as repeating and temporally coordinated activity of neurons. However, understanding of their role in brain function is hampered by a lack of scalable methods for their unsupervised detection. We propose using a graph neural network for embedding spike data into a sequence of fixed size vectors and clustering them based on their self-similarity across time. We validate our method on synthetic data and real neural recordings.

## 1 INTRODUCTION

Cell assemblies (Hebb, 1949) are observed as patterns of neural activity that repeat themselves with a high degree of similarity across time. However, their intrinsic variability (inexact relative timing, deletions or additions of spikes) makes it challenging to detect them without a behavioral reference (such as the position of the animal in space). While popular methods like PCA and ICA are certainly useful, they only capture temporally correlated activity of large groups of neurons but disregard the relative order of spikes. Attempting to overcome this limitation, a variety of methods have been proposed ranging from edit similarity-based template matching (Watanabe et al., 2019) and transport-based clustering (Grossberger et al., 2018) to convolutional non-negative matrix factorization (Mackevicius et al., 2019; Peter et al., 2016) and point process models (Williams et al., 2020). Unique from previous approaches, here we treat neural data as a sequence of directed weighted graphs, each of which is transformed with a graph neural network into fixed-size embeddings and clustered using a standard clustering algorithm. By representing data as graphs, our method leverages the natural sparsity of neural activity, and should scale well to very large datasets. We demonstrate the performance of the method on synthetic and real data.

## 2 METHOD

*Data preparation.* We start with a recording of $N$ neurons for $T$ time steps, $\mathbf{X} \in \{0,1\}^{N \times T}$, such that $X_{n,t} = 1$ if there is a spike at time $t$ on the $n$-th neuron and $X_{n,t} = 0$ otherwise. This matrix is then segmented into $M$ overlapping windows $G^{(t)} = \mathbf{X}_{:,t:t+w}$, $t \in \{0, k, 2k, \ldots, M\}$, $M = T//k$, $w = 200$ is the window length in samples and $k = 4$ is the number of time steps between two adjacent windows. Each of the windows is converted into a weighted directed graph $\mathcal{G}^{(t)}$, in which the nodes represent spikes, and the edge weights from the $i$-th to the $j$-th neuron are a function of time difference $\Delta t$ between two consecutive spikes on those neurons. In this paper we calculated edge weights as $e^{-\Delta t/\tau}$, with $\tau = 25$, to emphasize the contribution of spikes that occur in close temporal proximity. For any two spikes on neurons $i$ and $j$, the edge weight is non-zero if $t_i < t_j$. If more than two spikes occur between the same pair of neurons, the weight of the edge between them is the sum of the contributions of all the spikes within that window. The nodes (neuron indices) are encoded as integers $\{0, \ldots, N-1\}$.

*Model.* We use a stack of 3 graph convolution layers (Kipf & Welling, 2016) with output dimensions of 10, 10 and 6, respectively, each followed by an ELU non-linearity. Neuron indices are converted

to 10-D vectors using an embedding layer. Since each window of the original data contains a varying number of spikes (hence a different number of nodes in the corresponding graph), we use a global average pooling layer on top of the GCN stack to obtain a fixed size embedding for each graph.

*Training*. We fit the model using the AdamW optimizer with default parameters to minimize the following loss function: $\mathcal{L}(\theta) = XE(\mathbf{Z}, \mathbf{y}) + \beta TV(\mathbf{Z})$, where $XE(\mathbf{Z}, \mathbf{y})$ is the cross-entropy loss between the embeddings and their cluster assignments (recalculated at the beginning of each epoch using K-means ($K = 6$) as in (Caron et al., 2018; Hsu et al., 2021)) and $TV(\mathbf{Z}) = \sum_{i=1}^{6} \sum_{t=1}^{M-1} (Z_{t,i} - Z_{t+1,i})^2$ is the total variation of the embeddings over time, which encourages temporal consistency of the class labels. Using K-means cluster assignments as targets guides the optimization towards a representation in which similar patterns are close, while different ones are well separated. We set $\beta$ to 0.1 and continue training until the cross-entropy loss plateaus.

## 3 RESULTS

*Synthetic data*. To facilitate fair comparison of the model's performance on real and synthetic data, the synthetic dataset was made to match the properties of the real CA1 recording we consider in the next subsection. We embedded 3 artificial spike patterns into a binary matrix of background activity $\mathbf{X} \in \{0, 1\}^{N \times T}$ ($N$=452, $T$=18137) obtained by permuting inter-spike intervals of the CA1 dataset (ensuring that the synthetic and real datasets had approximately the same spike statistics). Each of the artificial patterns was a temporally jittered (std=10 time steps) sequence of 100 spikes: one spike per neuron per time step. The patterns were pruned by dropping spikes with a probability of 0.2.

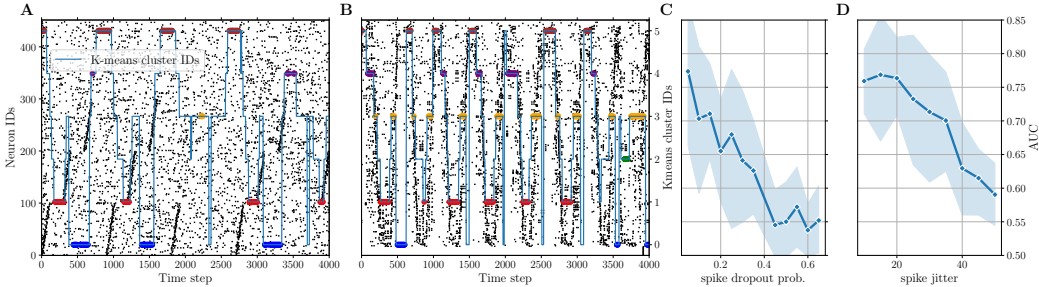

Figure 1: The presence of self-consistent clusters (colored dots) indicates statistically significant patterns. The detections roughly correspond to patterns in synthetic data (A) and similar portions of place cell sequences in real CA1 data (B). The detections appear earlier than the corresponding patterns because $\mathcal{G}_t$ is computed from $\mathbf{X}_{:,t:t+w}$. For details on significance testing, see Appendix A. The neurons are sorted to reveal the patterns, but this is not required for the method to work. (B) and (C) show the models performance as a function of temporal spike jitter and spike dropout, respectively.

*Real data*. We used data from (Rubin et al., 2019) recorded from the area CA1 of a mouse running on a linear track to collect water rewards. With the same hyperparameters as for the synthetic data, the model was able to distinguish sequential activity of place cells (Fig 1A) with similar spatial tuning (although with a lower accuracy, which is likely due to the animal's non-constant speed on the track).

## 4 CONCLUSIONS AND FUTURE DIRECTIONS

We proposed a method for unsupervised detection of patterned neural activity, which represents spike data as a sequence of directed weighted graphs. Given the natural sparsity of cortical neurons, this approach might offer superior performance and enhanced sensitivity compared to other methods. However, establishing whether that is the case requires rigorous speed benchmarking, which we leave to future work. The code is available at https://github.com/RomanKoshkin/GraphEx.

URM STATEMENT

The authors acknowledge that at least one of the authors of this work meets the URM criteria of ICLR 2023 Tiny Papers Track.

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

## A  STATISTICAL TESTING

*Statistical testing.* We consider a pattern as statistically significant if it is self-consistent across 90 % of model instantiations (each of which initialized with the same hyperparameters, but different weights and cluster centroids, and trained for the same number of epochs on the same data) (Fig. 2, 3 and 4). The self-consistency here means that strong patterns will produce very similar cluster transitions over time (although their integer labels are not guaranteed to be the same). To clarify this further, consider the case with one expected pattern (we set $K = 2$ to allow one extra cluster for background activity). We train the model several times, and for each training run we obtain a sequence embeddings $\{\mathbf{e}_t\} \in \mathbb{R}^2$ and their corresponding cluster assignments $\{c_t\} \in \{0, 1\}$, $t \in \{1, \ldots, T\}$. Then, using one of the embedding vectors from the first run as a reference, we select embedding vectors that have the highest correlation with the reference one. Finally, we stack the vectors of cluster assignments corresponding to the embedding vectors selected before and average them over the model runs. The resulting average vector provides a measure of consistency of patterns detected in different model runs (Fig. 4, 3). Conveniently, the model's relatively small size (4842

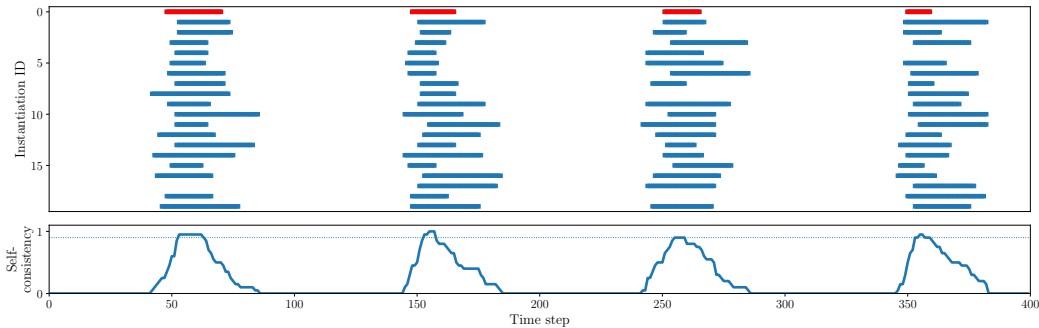

Figure 2: Schematic illustration of the procedure to test pattern significance. Top: binary matrix each row of which indicates the time steps at which the model assigned the corresponding graph to the cluster that happened to have the highest similarity with the reference (top row, red). The average over the rows of that matrix (bottom) provides a measure of self-consistency of the reference pattern. Dotted line marks the significance criterion.

parameters) enables training dozens of its instantiations quite quickly on a modern GPU (training for 150 epochs, which is sufficient for convergence, epochs takes about 40 s on an NVIDIA RTX A6000). The self-consistency criterion we enforce in this paper (90 %) can be relaxed depending on the nature of data and the level of confidence desired.

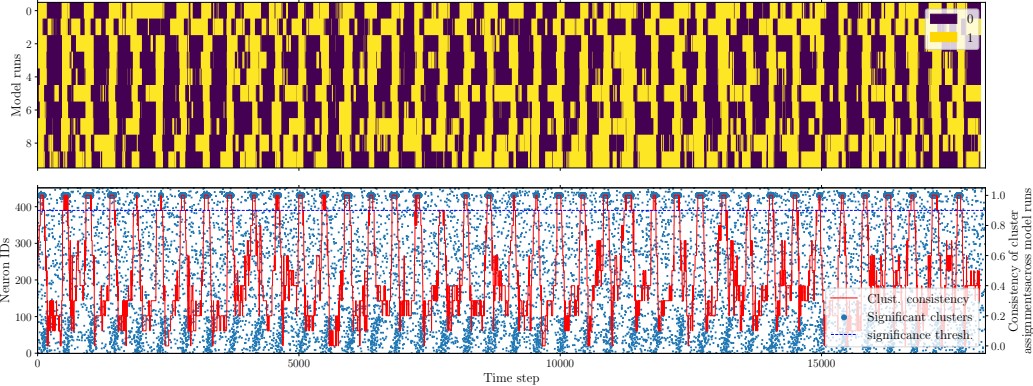

Figure 3: Illustration of the statistical testing procedure. Top: final cluster assignments in 10 model instantiations. Reference cluster assignments (Run 0) are in the top row. Bottom: training spike data, average cluster assignments over model runs and statistical significance threshold (horizontal dashed line). Significant pattern detections are marked with blue round markers.

## B  DETECTING PATTERNS IN SONGBIRD DATA

Additionally we tested our method on the songbird HVC dataset from Williams et al. (2020), and for comparison we also provide the same results obtained with *PP-Seq* (Fig. 7) and *seqNMF* (Fig. 6).

## C  SPEED COMPARISONS

We compared run times of our model to those of two other models (Table 1). Our model was fit for 150 epochs on a dataset of 452 neurons and 18137 timesteps with sequences of 100 neurons embedded every 450 timesteps (with a jitter of 10 timesteps and a spike dropout probability of 0.2). The *seqNMF* and *PP-seq* were fit with default parameters on the same data. For all the three models we used the same machine (Xeon Gold 6226R CPU @ 2.90GHz with 32 cores, 96GB of RAM).

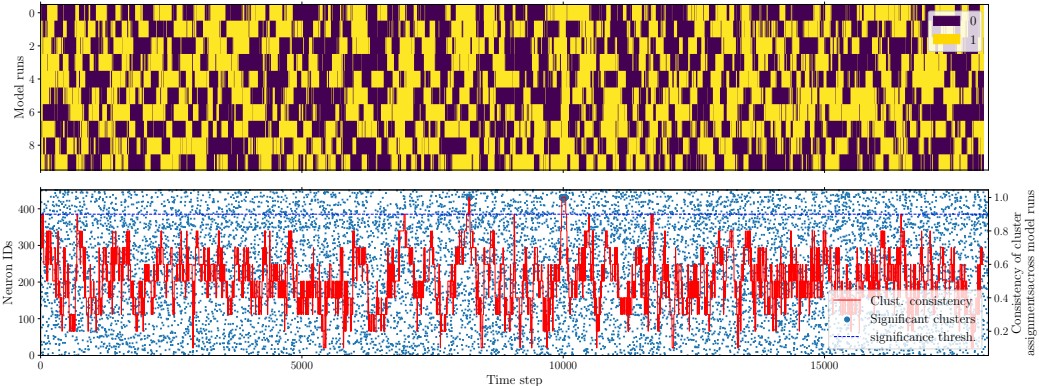

Figure 4: Same as above, but with null data (i. e. containing no patterns). Notice the two spurious pattern detections suggesting that the significance threshold of 90% should be increased.

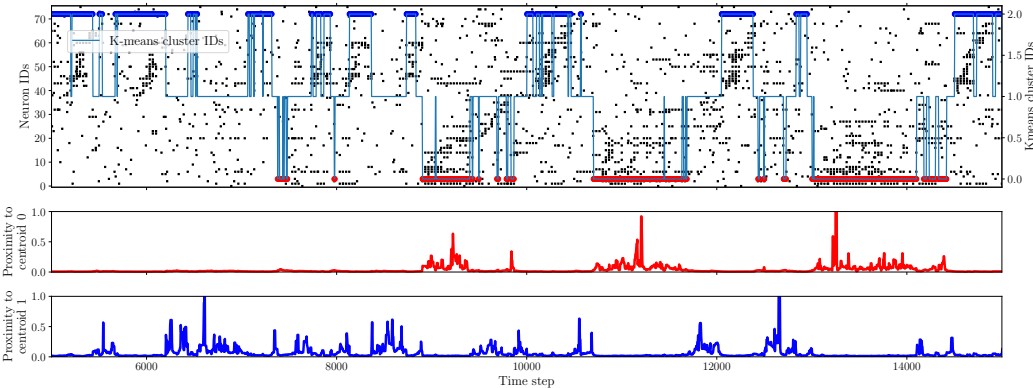

Figure 5: The songbird dataset contains two patterns. Our model was able to determine the times at which the patters were expressed.

Table 1: Speed comparisons. Standard deviations over 10 runs are in brackets.

| Method | Run time |
|--------|----------|
| Ours | 39.3 (0.32) s |
| PP-Seq (Williams et al., 2020) | 39.3 (0.14) s |
| seqNMF (Mackevicius et al., 2019) | 1058 (9.8) s |

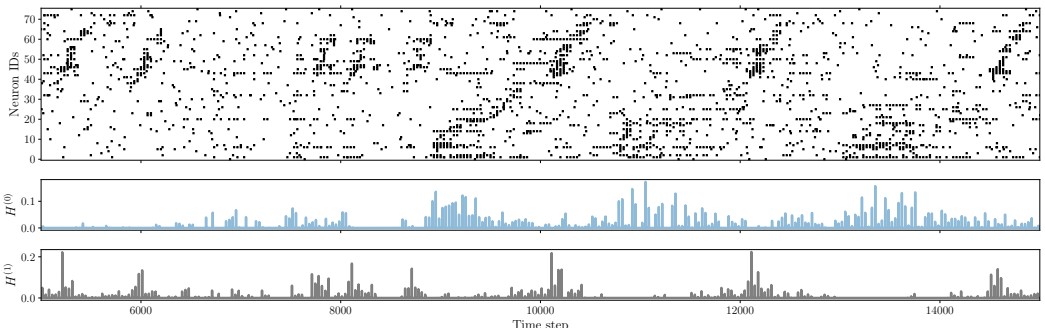

Figure 6: Same as Fig. 5, *seqNMF*. Bottom panels: temporal loadings of the two patterns.

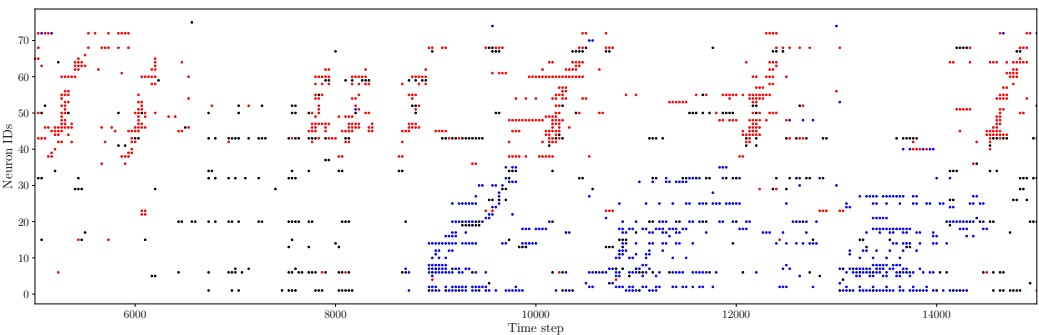

Figure 7: Same as Fig. 5, *PP-Seq*. Colors indicate pattern assignments.

