# OpenReview forum: "Unsupervised Detection of Cell Assemblies with Graph Neural Networks"
_ICLR.cc/2023/TinyPapers — Submitted to Tiny Papers @ ICLR 2023_

### Official Review · Reviewer_79tn · 2023-03-26

**Confidence:** 4

**Summary Of Contributions:**

This paper proposes a GNN-based unsupervised embedding method for the detection of cell assemblies in spike recordings.

**Rating:**

Great Start (GS): a submission which meets some of the reviewing criteria but has room for improvement

**Strengths And Weaknesses:**

I assess the submission with respect to each of the reviewing criteria below:

- **Clarity**: The writing is generally clear. However, there is room to improve the description of the method, and the significance testing approach is not clearly motivated (see **Suggested Changes**).

- **Correctness**: The experiments appear to be correctly performed, but are limited to a single dataset. Benchmarking the proposed algorithm against existing methods would make for a stronger paper. Admittedly, the authors explicitly state that they leave this for future work, but without such comparisons it is not clear whether the proposed method represents progress towards solving the motivating problem. See **Suggested Changes** for more details.

- **Reproducibility**: The experimental setup is described fairly clearly, but code is not made available.

- **Follows basic requirements**: The paper fits within the length requirement, and is adequately anonymized.

**Suggested Changes:**

**Major**:

1. Some of the design decisions should be more clearly motivated. For example, why is the time constant in the edge weighting set to $\tau = 0.04$? In what units is this measured? If the goal is to "emphasize the contribution of spikes that occur in close temporal proximity" (Lines 31-32), shouldn't this parameter be adjusted depending on the acquisition rate of the underlying recording (if it's measured in units of data samples) and on the relevant timescale over which activity in the area of interest evolves?

2. I think it would be useful to more clearly motivate the GNN model. Some of the prior works cited (e.g., Williams et al.) build up their methods from clearly-defined statistical models. Though the present method is not set up in that way, it would be useful to motivate why graph encoding and GNN embedding is a sensible way to detect structure in spike data that would result from the existence of assemblies. In the submitted manuscript, the paper simply states the algorithm without motivating its design.

3. As mentioned above under **Correctness**, I find it hard to judge the usefulness of the method vis-à-vis the problem of assembly detection stated in the introduction without any explicit comparisons against previously-proposed methods. The experiment on place cell recordings in mice running on linear tracks is nice, but one can detect the sequential structure present there just from firing rates. In this case, also, the clustering shown in Figure 1 seems to artificially discretize the population into clusters that respond within a certain interval of the track. It would be useful to provide a direct comparison of the proposed method with at least one of the previously-proposed algorithms on a more complex dataset that contains more clearly discretized syllables, e.g. the zebra finch HVC data considered in Williams et al. and Mackevicius et al.

4. The criterion for statistical significance of a pattern needs to be justified more rigorously. As described, it is as much an assessment of the robustness of model fitting as an assessment of characteristics of the pattern. At the very least, I would suggest that the authors benchmark this approach on toy synthetic data containing very clear clusters, to give a baseline measure of how variable the clustering is for noiseless data. Also, they should apply the method to null data containing no clusters, and show that this approach successfully fails to reject the hypothesis that there are no clusters present.

**Minor**:

1. In the top panel of Figure 2, the ordinate axis label is misspelled: "Instantion" -> "Instantiation."

---

> ### Author Response · Authors · 2023-05-31
> **Response to Reviewer 79tn**
>
> Thank you very much for the careful review of our work and your helpful feedback. Below are our responses to the points raised.
>
> - Why is the time constant in the edge weighting set to $\tau = 0.04$? In what units is it measured?
>
> Thank you for raising this point. As reviewer by the other reviewer suggested, we changed the weighting of the edge weights from $\exp(-\Delta t * \tau)$ to $\exp(-\Delta t / \tau)$ in line with the intuition that larger values of \tau should reflect longer decay of the edge weight as a function of $\Delta t$. The particular choice of value 0.04 (now changed to 25 to ensure the same edge weighting) is empirical and we found it to work well on the CA1 dataset as well as the synthetic datasets whose spike statistics were made to match that of the CA1 dataset. Although we did not perform comprehensive analysis of how $\tau$ affects our model’s performance, we note that its effect is quite small (except with very fast $\tau$, in which case the graph has a very short time “horizon”). The choice of $\tau$ is closely related to the window size: choosing a fast $\tau$ would be wasteful with a long window as distant spikes would have almost zero contribution to the graph overall. $\tau$’s unit of measurement is time samples / ln(2). A good intuition for choosing a suitable value of $\tau$ would be to set it such the weight “halflife” (i. e. the number samples between the spikes at which the edge weight would 0.5) is some fraction (e. g. 1/6) of the window length. For example, if the window is 100 time steps, then a sensible value for $\tau$ would be $100/6 / \ln(2) \approx 24.04$. We set \tau guided by this logic to the round number 25.
>
> - it would be useful to motivate why graph encoding and GNN embedding is a sensible way to detect structure in spike data
>
> It is believed that spiking activity has a graph structure because it is generated by a neural network, which itself is a graph. Also, spiking activity is quite sparse. Convolution-based methods (e.g. convNMF, seqNMF), requires representing spiking data as a dense binary matrix, most elements of which are zeros. In fact, in the datasets we used in our paper (including the real CA1 recording from the mouse hippocampus), about 99.85% of the elements in the data matrix (out of which we construct graphs) are zeros. By convolving a filter with the data, most multiplication operations will involve wasteful multiplications by zeros. A graph representation of data is more efficient given the sparsity of spiking activity. To make this point more concrete, we compared the run times of our method with PP-Seq and seqNMF and provide results in Table 1.
>
> - I find it hard to judge the usefulness of the method vis-à-vis the problem of assembly detection stated in the introduction without any explicit comparisons against previously-proposed methods.
>
> Due to the page limit, we did not provide detailed comparisons with other methods. However, ours performs comparably to others on the datasets we tested. In the revised manuscript, we added two panels showing how the model performs, as well as additional Figures in Appendix B, where we compare our method with seqNMF and PP-Seq in the ability to detect two patterns in the HVC dataset (used in Williams et al. 2020).
>
> - one can detect the sequential structure present there just from firing rates.
>
> The idea of the method is to detect sequential structure, not simply periods of increased correlated firing. If the patterns only constituted correlated activity in which the order of the spikes does not matter, then detecting such (non-sequential) patterns would be easy with methods such as PCA or ICA. Our method, as do PP-Seq, convNMF and seqNMF) aims to detect sequential patterns.
>
> - it would be useful to provide a direct comparison of the proposed method with at least one of the previously-proposed algorithms on a more complex dataset that contains more clearly discretized syllables, e.g. the zebra finch HVC data considered in Williams et al. and Mackevicius et al.
>
> We have added comparison of our method with convNMF and PP-Seq on the HVC data in Appendix B.
>
> - I would suggest that the authors benchmark this approach on toy synthetic data containing very clear clusters… Also, they should apply the method to null data containing no clusters, and show that this approach successfully fails to reject the hypothesis that there are no clusters present.
>
> Thank you for your suggestion. Appendix A now contains the demonstration of how method’s performs on null data and data containing patterns.
>
> - In the top panel of Fig. 2, the ordinate axis label is misspelled
>
> Thank you for spotting that. This is now fixed.

---

### Official Review · Reviewer_R8V8 · 2023-03-28

**Confidence:** 4

**Summary Of Contributions:**

This paper proposes a method of embedding spike data into clusters based on their self similarity across time. First a time-varying directed graph between neurons is constructed with edge weights based on how close one spike follows another. Then a GNN is applied at each time to (soft) assign each time window to a cluster.

**Rating:**

Clear, Correct, and Reproducible (CCR): a submission which meets the reviewing criteria

**Strengths And Weaknesses:**

Context: I have little experience in this datatype, but significant experience in GNNs and clustering methods. Cool work! Thanks for the read.

Strengths:

- The model specification was quite clear and reproducible.
- The losses seem reasonable.
- I think this is a cool idea, essentially this is a problem of clustering graph structures (with the same number of nodes). I haven’t seen this problem setup before. I’m unclear if a GNN is necessary here, but certainly an interesting idea to try.

Weaknesses:

- The results are qualitative up to this point.
- Its unclear from the exposition exactly why we want to cluster time periods of the neurons. “The model successfully disentangles repeating patterns in both the synthetic and real datasets”. This seems interesting, but it would be nice to add a sentence on why this is interesting / useful. I assume they correlate with the mouse doing something maybe?
- Why not include all of the neurons in the graph so that the graph size is fixed? (There may be some nodes with no connections but that is okay). This is one of the points I didn’t get.

**Suggested Changes:**

Suggested Changes:

- Add more context on why we want to cluster neuron activity patterns over time.
- I didn’t quite understand why things were updated at the beginning / end of each epoch and what effect this has. Perhaps this could be clarified in a sentence?

(Possibly) interesting ideas:

- You may want to add additional features (besides the neuron embeddings) to the input features of the GNN. Work on molecular graphs has found degree, and k-eigenvectors as useful additional features.
- Since the graph is directed, I would be surprised if GCN works very well as compared to more recent GNN methods. Ideas around the magnetic Laplacian could be of interest here.
- Obviously (as mentioned by the authors) more work in benchmarking this method would be interesting too.

---

> ### Author Response · Authors · 2023-05-31
> **Response to Reviewer R8V8**
>
> - Cool work! Thanks for the read!
>
> Thank you very much for the positive evaluation of our work
>
> - it would be nice to add a sentence on why this is interesting / useful.
>
> The problem of unsupervised pattern detection is relevant and interesting because of the need to study structured spontaneous neural activity, which is believed to be crucial in information processing during sleep. When a mouse runs of a track, we can record the activity of its neurons simultaneously with its location in space (and other information that might be used as a reference for detecting patterns later). In the case of hippocampus, we can determine in which place(s) on the track each recorded neuron fired most and then sort these neurons into their “preferred space bins”, aka place fields. Knowing the neurons’ place fields makes it easy to detect “replay events” in a mouse that is immobile (e. g. during sleep). It is much more interesting to detect repeating patterns of neural activity for cells other than those encoding spatial location, because structured spontaneous activity is hypothesized to play a crucial role in neural information processing. Detecting patterned neural activity for which no behavioral reference is available requires unsupervised pattern detection methods.
>
> - Why not include all of the neurons in the graph so that the graph size is fixed?
>
> Thank you for this idea. We excluded non-spiking neurons based on the logic that they should not contribute to computing the output embedding. In the interests of time, we leave exploring this to future work.
>
> - I didn’t quite understand why things were updated at the beginning / end of each epoch and what effect this has. Perhaps this could be clarified in a sentence?
>
> Before a forward and backward pass, we update the centroids of the K clusters.  We then perform a forward and backward pass on the whole dataset (one epoch) to update the weights in a way to move the embeddings closer to their respective centroids. At convergence (which usually takes about 100-200 epochs with the hyperparameters we used), the cross-entropy loss plateaus and the cluster centroids stabilize. This signals that further training is not needed. We added a sentence to the revised manuscript to make this clearer (lines 47-49).

---

### Official Review · Reviewer_3TSU · 2023-03-30

**Confidence:** 3

**Summary Of Contributions:**

Cell assemblies are units of neural computation that manifest themselves as repeating and temporally coordinated activity of neurons. In this paper, the authors propose using a graph neural network to embed spike data into a sequence of fixed-size vectors and clustering them based on their self-similarity across time. They validate their method on synthetic data and real neural recordings.

**Rating:**

Great Start (GS): a submission which meets some of the reviewing criteria but has room for improvement

**Strengths And Weaknesses:**

# Strengths
1. This paper designs a method for cell assemblies detection based on graph neural network, which overcomes the problems of traditional methods such as template matching and non-negative matrix factorization.
2. This paper designs an unsupervised training goal, that is, to ensure the consistency of the same spike embedding in different time windows, and shows the effectiveness of their proposed method on two data sets.

# Weaknesses
1. It is not a novel idea to encode biological data into a graph and use graph neural network to mine the relationship on the graph. I think the author needs to further clarify the motivation.
2. As the author mentioned, this paper does not give a specific quantitative experimental measurement metric, and there is also a lack of comparative experiments with baseline methods.
3. Both datasets are relatively small in size, and it is doubtful whether the proposed method can perform the same on larger datasets.


**Suggested Changes:**

In general, I think this paper is a good start, and the author needs to comprehensively consider the points mentioned in the *Weaknesses* to make further improvements to the paper

---

> ### Author Response · Authors · 2023-05-31
> **Response to Reviewer 3TSU**
>
> Thank you very much for the careful review of our paper and your helpful comments.
>
> - I think the author needs to further clarify the motivation (for representing data as graphs)
>
> Spiking activity is quite sparse. Convolution-based methods (e.g. convNMF, seqNMF), requires representing spiking data as a dense binary matrix, most elements of which are zeros. In fact, in the datasets we used in our paper (including the real CA1 recording from the mouse hippocampus), about 99.85% of the elements in the data matrix (out of which we construct graphs) are zeros. By convolving a filter with the data, most multiplication operations will involve wasteful multiplications by zeros. A graph representation of data is more efficient given the sparsity of spiking activity. To make this point more concrete, we compared the run times of our method with PP-Seq and seqNMF and provide results in Table 1.
>
> - Lack of comparative experiments with baseline methods.
>
> We added basic performance benchmarks to Fig. 1 and speed comparison in Table 1 (in Appendix).
>
> - Both datasets are relatively small in size, and it is doubtful whether the proposed method can perform the same on larger datasets.
>
> In terms of speed, our method is faster than seqNMF (we added a brief comparison in Table 1 in the Appendix).

---

### Comment · Area_Chair_hSro · 2023-06-06
**Meta-reviewer final comment**

Thank you for very diligently responding to all the raised points, resulting in a great TinyPaper.
This work meets the threshold for archival, contents the URM statement and is deanonymized

---

### Meta-Review · Area_Chair_hSro · 2023-04-03

**Recommendation:** Invite to archive
**Confidence:** 4

**Metareview:**

Thank you for this interesting paper!

This paper make a novel proposal for how to do unsupervised detection of neural assemblies. It is significant as it paves a way for graph neural networks to be used to efficiently analyse neural responses from unlabelled spiking data alone. The paper is well written and the model and methods are clearly explained however the results are less clearly explained, to the extent that it is a little hard to judge how well the model performs.

To the best of my knowledge I have not seen this approach to solving the ensemble-finding problem and the initial work here, although incomplete, contains valuable contributions and may prompt interesting future work. The reviewers are broadly in agreement about the following pros and cons

Pros:
* A well explained model, including clear details of how the data is preprocessed.

Cons
* The results are somewhat unconvincing; lacking benchmarking or objective measures of perfromance.

I have meta-reviewed this as "Invite to archive". In the space of two pages the authors clearly present an interesting problem, a novel solution including methods, and some positive (if improvable) initial results. Some further work would be needed to convince readers of its performance and thus get a higher recommendation.

**Summary:**

This paper presents a novel technique for unsupervised spike sequence (or "neural ensemble") detection which is significantly different to other proposed techniques. It is based on graph neural networks, thus can effectively exploit the sparse and network-based origins on the model. The main weakness, as pointed out by all reviewers, is that it is unclear how well this model performs both in absolute terms and relative to comparable techniques.

**Comments And Feedback To The Authors:**

## Minor revisions to be added:

Please implement the suggested changes of reviewers in to the manuscript. Could you please also clarify in one sentence how the K-means cluster provides an independent target for the optimizer. I would also be essential for you to explain more clear why figure 1 shows your model is working. Of course it is apparent there is correlation between the K-means cluster labels and the ground truth embeddings (e.g. in the synthetic example) but it's not one-to-one.

Additionally some minor errors need fixing:

* Is there a typo on line 31 and it should read: $e^{−\frac{∆t}{\tau}}$
* It seems incorrect to describe Williams (2020) as a non-neg matrix factorisation technique like, say, Mackevicius (2019). Please update this.
* I agree with reviewer R8V8 that you should at least try to include the non-spiking neurons. It's strange they're taken out artificially forcing you to use a pooling layer.

## Major revisions:

I will not ask the authors to perform benchmarking against other techniques due to time limitations although that would be nice. I will leave it up to the authors, but strongly encourage them, to add one more panel to figure one with some form of easy-to-understand objective metric for how the model is performing.

If a significant number of these revisions are satisfied and the results are provably good I would consider increasing this recommendation to "invite to present".


**Reason For Not Giving A Higher Recommendation:**

Two of three reviews rated this paper a "great start". I agree with this judgement. It's a very promising idea.

Unfortunately the lack of a quantitative measure of performance (and therefore no direct apples-to-apples comparisons to say, Mackevicius (2019) or Williams (2020)) unfortunately makes it hard to know just how good this technique is. This is a shame as the idea of decoding neural ensembles with GNN seems pretty reasonable.

The reviewers seem a little unconvinced with the results, which seems fair. It is also unclear to me how the results show "good" clustering. For example the K-means labels don't overlap perfectly with the locations of neural ensemble embeddings in figure 1 (left). Why is this? The model seems to decode sequences in regions outside sequence embeddings, presumable this is due to the large window length. If it was expected that the model should not uniquely select for individual embeddings/ensembles then this could be explained better.


**Reason For Not Giving A Lower Recommendation:**

In the space of a two-page paper it is unreasonable to expect the authors to do significant benchmarking (but a minimal amount would be required for higher score). The fact that the datasets used are only small does not rule out the technique from scaling -- perhaps with modifications -- in future work.

Although as reviewer 3TSU says "it is not a novel idea to encode biological data into a graph" that is not at all a flaw and is, actually, what motivates the study since biological data originates from structures with graph-like architectures. To the best of my knowledge I have not seen this approach to solving this problem and the initial work here, although not finished, contains value.

---

> ### Author Response · Authors · 2023-05-31
> **Response to Area Chair hSro**
>
> Thank you for your very helpful and diligent review. We answer the reviewer's specific comments below.
>
> - ...lacking benchmarking or objective measures of performance; unclear how well this model performs both in absolute terms and relative to comparable techniques; lack of a quantitative measure of performance (and therefore no direct apples-to-apples comparisons to say, Mackevicius (2019) or Williams (2020)).
>
> Thank you very much for pointing out the omission. We have now added basic performance measures (AUC) to Fig. 1 of the paper and additional data in Appendix B. We followed a similar approach to Williams et al. (2020) and tested how the model’s ability to detect one pattern depends on spike dropout probability and spike timing jitter (which reflects how far (in time steps) the timing of spikes participating in a pattern can deviate from their ideal sequential timing). The spikes were jittered by adding random time shifts $t_{disp} \thicksim \mathcal{N}(0, \sigma_{jitter})$ to the original pattern.
>
>
> - the K-means labels don't overlap perfectly with the locations of neural ensemble embeddings in figure 1 (left). Why is this?
>  - it is apparent there is correlation between the K-means cluster labels and the ground truth embeddings (e.g. in the synthetic example) but it's not one-to-one.
> - The model seems to decode sequences in regions outside sequence embeddings, presumable this is due to the large window length. If it was expected that the model should not uniquely select for individual embeddings/ensembles then this could be explained better.
>
> Thank you very much for this question. The K-means cluster assignments in Fig. 1 indeed appear to precede the actual times at which the patterns actually occur in the data, and they precede them by approximately w. This results from the fact that a cluster assignment at time t is computed for the window that starts at t and ends and t + w. The revised manuscript clarifies this in the caption of Fig. 1. Also, the decoding performance of our model (as perhaps any other similar model) is not perfect, and in general some errors are to be expected especially when the pattern is weakly expressed or distorted (due to dropout and/or spike timing jitter) or obscured by background noise. Indeed, the choice of window length affects the decoding performance. Specifically, long windows can capture longer patterns without fracturing them into what would appear as multiple patterns (but in fact are one long pattern), but might not detect smaller patterns.
>
> - Clarify in one sentence how the K-means cluster provides an independent target for the optimizer.
>
> If the time-series data contains repeating patterns, the cloud of data points already have regions of relatively high density (that is clusters or relatively similar points), and even before the model optimization, the K-means algorithm is more to “pick up” these regions than others. However, the initial clusters are usually not well separated and the datapoints need to be embedded/transformed so that the clusters (self-similar datapoints) become well defined. Our model is trained (intermittent with cluster centroid updates) essentially to ensure maximum separation between clusters and dense packing of the data points (graph embeddings) within clusters. This works thanks to the cross-entropy loss which encourages the embedded datapoints assigned to one cluster to move closer to its corresponding centroid. On the other hand, the data that contains no pattern is spherical (that is without regions of high density), and although our model will still be able to find spurious patterns in such data, their corresponding clusters will not be reproducible across multiple runs (given a random seed). More precisely, the across-run correlation between embeddings will be low for a dataset that contains no patterns and relatively high for data that does have patterns. We hope this is now more clear in the manuscript (lines 47-49).
>
> - Explain more clearly why Fig. 1 shows your model is working.
>
> We hope that the caption to Fig. 1 now explains this succinctly and clearly.
>
> - Is there a typo on line 31?
>
> Thank you for pointing this out. This was not a typo, but indeed, for $\tau$ to be interpreted as a decay constant of the weight as a function of distance between the spikes, it should be in the denominator of the power. After fixing that, the value of $\tau$ is changed to from 0.04 to 1/0.04=25 (to keep this hyperparameter effectively unchanged).
>
> - It seems incorrect to describe Williams (2020) as a non-neg matrix factorisation technique like, say, Mackevicius (2019).
>
> Thank you very much for pointing this out. We fixed that.

---

> > ### Author Response · Authors · 2023-05-31
> > **Response to Area Chair hSro**
> >
> > - I agree with reviewer R8V8 that you should at least try to include the non-spiking neurons. It's strange they're taken out artificially forcing you to use a pooling layer.
> >
> > Thank you for this idea. We initially excluded non-spiking neurons based on the logic that they should not contribute to computing the output embedding. In the interests of time, we leave exploring this to future work.
> >
> > - I will not ask the authors to perform benchmarking against other techniques due to time limitations although that would be nice. I will leave it up to the authors, but strongly encourage them, to add one more panel to figure one with some form of easy-to-understand objective metric for how the model is performing.
> >
> > We have added basic performance metrics (AUC) to Fig. 1, as well as an illustration of how our model detects patterns in the HVC data (as suggested by reviewer 79tn) compared to seqNMF and PP-Seq.

---

### Decision · Program_Chairs · 2023-04-10

Invite to archive

---

> ### Author Response · Authors · 2023-05-31
> **Response to Paper Decision**
>
> Thank you very much. We wish to opt in for paper archival.